# 3D-Printed Medical-Grade Polycaprolactone (mPCL) Scaffold for the Surgical Treatment of Vaginal Prolapse and Abdominal Hernias

**DOI:** 10.3390/bioengineering10111242

**Published:** 2023-10-24

**Authors:** Mairim Russo Serafini, Alexandra Mowat, Susanah Mustafa, Siamak Saifzadeh, Tara Shabab, Onur Bas, Nicholas O’Rourke, Dietmar W. Hutmacher, Flavia Medeiros Savi

**Affiliations:** 1Department of Pharmacy, Universidade Federal de Sergipe, São Cristóvão 49100-000, Brazil; maiserafini@hotmail.com; 2Centre in Regenerative Medicine, Faculty of Engineering, Queensland University of Technology, Brisbane, QLD 4059, Australia; siamak.saifzadeh@qut.edu.au (S.S.); shbtara@gmail.com (T.S.); onur.bas@health.qld.gov.au (O.B.); dietmar.hutmacher@qut.edu.au (D.W.H.); 3Faculty of Medicine, University of Queensland, Brisbane, QLD 4072, Australia; 4Queen Elisabeth II Jubilee Hospital, Brisbane, QLD 4108, Australia; mustafa.susana@gmail.com; 5Medical Engineering Research Facility, Queensland University of Technology, Brisbane, QLD 4032, Australia; 6Australian Research Council Industrial Transformation Training Centre in Additive Biomanufacturing, Queensland University of Technology, Brisbane, QLD 4059, Australia; 7Department of Hepato-Pancreato-Biliary Surgery, Royal Brisbane and Women’s Hospital, University of Queensland, Brisbane, QLD 4029, Australia; orourke.nick@gmail.com; 8ARC Training Centre for Multiscale 3D Imaging, Modelling and Manufacturing, Brisbane, QLD 4059, Australia

**Keywords:** pelvic floor prolapse, mesh, mPCL, polypropylene, 3D printing

## Abstract

The expected outcome after a scaffold augmented hernia repair is the regeneration of a tissue composition strong enough to sustain biomechanical function over long periods. It is hypothesised that melt electrowriting (MEW) medical-grade polycaprolactone (mPCL) scaffolds loaded with platelet-rich plasma (PRP) will enhance soft tissue regeneration in fascial defects in abdominal and vaginal sheep models. A pre-clinical evaluation of vaginal and abdominal hernia reconstruction using mPCL mesh scaffolds and polypropylene (PP) meshes was undertaken using an ovine model. Each sheep was implanted with both a PP mesh (control group), and a mPCL mesh loaded with PRP (experimental group) in both abdominal and vaginal sites. Mechanical properties of the tissue-mesh complexes were assessed with plunger tests. Tissue responses to the implanted meshes were evaluated via histology, immunohistochemistry and histomorphometry. At 6 months post-surgery, the mPCL mesh was less stiff than the PP mesh, but stiffer than the native tissue, while showing equitable collagen and vascular ingrowth when compared to PP mesh. The results of this pilot study were supportive of mPCL as a safe and effective biodegradable scaffold for hernia and vaginal prolapse repair, hence a full-scale long-term study (over 24–36 months) with an adequate sample size is recommended.

## 1. Introduction

Hernias of the anterior abdominal wall and vagina are common pathologies and remain a significant source of morbidity, partly due to the high recurrence rate with surgical native tissue repairs [1,2,3,4]. Polypropylene meshes have long been associated with high risks of infection, adhesion, erosion, and with significant patient morbidity, due to an unfavourable foreign body response, especially meshes for vaginal prolapse [5,6]. Coated polypropylene (PP) meshes were thought to improve abdominal and pelvic organ prolapse and overcome the risks of erosion via supporting adequate tissue integration and minimizing foreign body responses [7,8]. Despite promoting lesser adhesion by shielding the PP mesh from initial inflammatory response, adhesion formation was still observed following degradation of the absorbable coatings [9,10,11]. To address the erosion risks associated with pelvic organ prolapse, dermal grafts were also introduced to facilitate fibroblast infiltration and support. However, this type of mesh initially elicits a robust and rigid biomechanical response, also triggering a strong foreign body reaction and intense collagen deposition around the graft material, consequently, resulting in fibrous encapsulation of the material, which is closely linked to mesh exposure and persistent pain, as noted by Nolfi et al. [12].

Mesh-related complications are influenced not only by the chemical properties of the mesh used but also by the mesh architecture and its mechanical properties. A gradual increase in stiffness has been linked to time-dependent variations in biomechanical properties of PP meshes [13], likely due to incorporation of the mesh within the surrounding tissue over time. The large pore size of these meshes would induce cells to promptly synthesize collagen within the mesh threads [14,15], thus causing an increase in stiffness and decrease in mechanical strength, resulting in mesh contraction, exposure, and persistent pain [7,16,17].

Every year, about 40.000 hernia repair surgeries are performed in Australia and about 20 million repairs across the world [18]. In Australia, transvaginal meshes for vaginal prolapse were introduced in 2004 and, while some of the meshes were successful in treating particularly anterior wall prolapse, the complication rate, including pain and erosion into surrounding organs, was unacceptably high [19]. Some of these products have been removed from the market and international litigation is ongoing [20].

It is in this context that we decided to scientifically assess an alternative material, medical grade polycaprolactone (mPCL) as an alternative adjunct to vaginal and abdominal native tissue repair. This Food and Drug Administration (FDA)-approved material is a slowly absorbable polymer and acts as a scaffold for native tissue ingrowth, rather than as a permanent synthetic mesh. To date, studies in other body tissues (e.g., bones and the oesophagus) have demonstrated mPCL to be a promising platform for soft tissue support and regeneration [21,22,23,24].

mPCL has been used for more than four decades in a large number of FDA-approved medical devices [25,26]. The use of mPCL nanofibers has been shown to enhance biomechanical properties of regenerated tissue with maximal strength force comparable to PP meshes without increasing local stiffness [27,28]. Early in vivo evaluation of solution-electrospun mPCL composites for repairing hernias of the anterior abdominal wall have also resulted in soft tissue regrowth [28,29,30,31]. The use of a slow-degrading material such as mPCL, could potentially overcome the long-term fate of resorbable coatings and the use of xenografts, including loss of strength. To our knowledge, our research group is the first to describe and evaluate this innovative regenerative strategy in the vaginal environment of the sheep model.

In this study, we hypothesised that melt electrowriting (MEW) mPCL scaffolds loaded with platelet-rich plasma (PRP), will enhance soft tissue regeneration in abdominal and vaginal fascial defects in the sheep model.

## 2. Materials and Methods

### 2.1. mPCL Scaffold Preparation

mPCL mesh scaffolds were fabricated using MEW, a technology combining additive manufacturing and electrospinning principles, as described in detail elsewhere [32,33] (Appendix A). Briefly, mPCL pellets (Purasorb^®^ PC 12, Corbion Purac, Amsterdam, The Netherlands) were melted at a temperature of 85 °C in the print head of the system and extruded through a 23G needle using an electro-pneumatic pressure regulator (ITV1050, SMC, Tokyo, Japan) at an air pressure 2.5 bars. A high voltage of ~6 kV was applied to the molten polymer during its extrusion, which facilitated the formation of a fine polymeric jet. This polymeric jet was then 3D printed on a grounded collector in a layer-by-layer manner until achieving a mesh with a thickness of ~500 μm, which is similar to that of PP meshes (~400 μm), at a translational speed of 150 mm/min. During the printing process, the distance between the collector and needle was set to 4 mm.

The meshes were designed to mimic the complex biomechanical behaviour of the native soft tissue, such as anisotropy and non-linearity. To achieve this, fibres with curvy and straight architectures were used in the meshes, and different pore sizes (500 μm and 1000 μm) were employed for each direction (horizontal and vertical). The use of curvy fibres provided flexibility to one loading direction, whereas straight fibres provided firmness in the other loading direction, leading to mechanical anisotropy. Two different types of mesh scaffolds were fabricated for each implantation area (30 mm × 30 mm for the abdominal wall and 20 mm × 20 mm for the vaginal floor). The pore size of PP mesh was 2500 μm × 2500 μm and mPCL mesh was 500 μm × 1000 μm. Description of mesh sizes before and after implantation is provided in Appendix A.

### 2.2. Preparation of Platelet-Rich Plasma (PRP) and Loading of mPCL Mesh Scaffolds

PRP was obtained from each sheep by collecting blood in phlebotomy tubes containing sodium citrate anticoagulant. For each sheep, the blood tubes were centrifuged at 1000 RCF for 10 min to collect plasma following the second centrifuge at 3200 RCF for 10 min to collect PRP. Next, 1 mL from the bottom of each tube was then collected and activated by 0.1 mL of CaCl_2_ (Calcium Chloride, 0.68 mmol/mL Ca, Phebra Pty Ltd., Lane Cove West, NSW, Australia). Each plasma-treated (O_2_/Ar_2_ plasma applied for each side for two min) and sterile scaffold (one scaffold for the abdominal wall and one scaffold for the vaginal floor for each sheep) was then covered with 0.5 mL of this mixture. The scaffolds were sterilised by immersing them in 80% ethanol for five min and then allowing the ethanol to evaporate in a biosafety cabinet overnight. The scaffolds were also exposed to UV light for 20 min. If the post treatment for surface hydrolysation and/or sterilisation is not performed according to protocol the mechanical properties of the scaffold can be reduced significantly (Appendix A).

### 2.3. Surgical Procedure

In this study, six parous ewes were allocated to two time-point groups of three months (n = 4) and six months (n = 2) (QUT Animal Ethics Approval Number 1800000002). The sheep were conditioned to QUT Medical Engineering Facility (MERF)’s customised sheep sling, 48 h prior to the experiments. Prior to the surgery, the animals underwent a single minor bleed (less than 7% of blood volume) to source PRP. Subsequently, animals were fully anaesthetised (Anaesthesia was induced with propofol (6 mg/kg bw, intravenous.) and maintained via mechanical ventilation with a mixture of oxygen, air and isoflurane (2%) and underwent surgical implantation of the two implants in the vaginal pelvic floor and abdominal wall (flank region). Each sheep received (i) PP mesh (Ethicon, Ultrapro^TM^ Monocryl^TM^ Prolene^TM^ Composite, Johnson & Johnson International) (control group), and (ii) mPCL mesh loaded with PRP (experimental group). Each sheep was implanted with two meshes in their abdominal wall, and two mesh scaffolds in their vaginal floor (Appendix A, respectively). The abdominal meshes were sutured into a surgically created square-shaped defect of 30 mm × 30 mm in the external oblique fascia, such that the peritoneal cavity was not perforated during mesh placement. The vaginal meshes were placed between the rectovaginal or rectovesical fascia and vaginal mucosa in surgically created square-shaped defects of 20 mm × 20 mm. The meshes were secured with minimal tension to the fascial layers in the defects with 2/0 polydioxanone (PDS) sutures placed at each corner of the mesh. Pre-emptive, intraoperative and postoperative analgesia were achieved with buprenorphine (0.005 mg/kg bw, intravenous), fentanyl infusion (2.5 mcg/kg bw/h, constant rate infusion) and fentanyl transdermal patch (0.5–1 mcg/kg bw/hr, q72hr), respectively.

After surgery, a urinary catheter and a vaginal pack were placed for 24 h, and an abdominal circumferential bandage was applied to support the surgical incisions. The animals were cradled in MERF’s customised sheep sling for 24 h post-surgery. The animals were housed indoors for a period of three weeks to ensure full healing of surgical incisions. Animals were monitored against defined clinical observations, three times per day for seven days after surgery, including weekends and public holidays, and once per day thereafter.

Upon completion of the experimental period, animals were humanely killed with intravenous Lethabarb^®^ (sodium pentobarbitone 325 mg/mL, Virbac Pty Limited, Brisbane, Australia, 100 mg/kg bw) and vaginal and abdominal tissues were collected for biomechanical, and histological analyses.

### 2.4. Mechanical Testing of Tissue–Mesh Complexes

Mechanical properties of the tissue–mesh complexes were assessed with plunger tests described in the reference [34], which are better at mimicking the physiological and biomechanical boundary conditions of the native tissue than uniaxial compression and tensile tests. This plunger test setup consists of a base specimen holder plate with a central circular opening, a second plate that is used for securing the specimens to the base plate, as well as a plunger with a spherical metallic tip attached to Instron 5848 Microtester equipped with 500 N load cell. During the tests, the samples were secured between the base and upper fixation plates with the use of bolts and nuts. Nails were placed around the specimens to prevent slippage. The plunger was pushed through the specimens at a displacement rate of 1 mm/s. A tare load of 0.05 N was applied for determining the starting point of the test during the analysis of the tests. This tare load excluded minor force readings arising from the surface features of the specimens and ensured that full contact between the plunger and samples was achieved. Two stiffness values were obtained for each specimen by calculating the slopes at the toe (the region between the displacement of 1 mm and 5 mm) and linear regions (the region between the displacement of 7.5 mm to 15 mm) of the force–displacement curves of the tested specimens. Some of the specimens had to be excluded due to the major irregularities in their shape, size and/or thickness.

### 2.5. Histology and Immunohistochemistry

#### 2.5.1. Histology

After biomechanical testing, samples were fixed in 4% paraformaldehyde (PFA, Sigma-Aldrich (Merck)–Melbourne, Australia, cat. no. 158127) for one week and transferred to 70% ethanol prior to tissue processing. Samples were processed using a tissue processor (Excelsior ES tissue processor, Thermo Fisher Scientific, Seventeen Mile Rocks, Brisbane, Australia, cat. no. ASHA78410023) overnight. Processed samples were then paraffin-embedded using an embedding station (Shandon Histocentre 3 embedding station, Thermo Fisher Scientific, cat. no. B64110040) and allowed to cool off. Paraffin-embedded blocks were then sectioned at 5 μm thickness using a microtome (Rotary microtome, Leica Biosystems, Nussloch, Germany, model no. RM2265, cat. no. 050338780). Paraffin ribbons were flattened in a 40 °C water bath and collected onto super frost polysine microscope slides (Thermo Fisher Scientific, Seventeen Mile Rocks, Brisbane, Australia cat. no. MEN SF41296PL). Paraffin slides were then dried at 60 °C for 16 h prior to histological and immunohistochemical staining.

Morphological tissue structure was evaluated via hematoxylin and eosin (H&E), Masson trichrome staining using a Leica Autostainer XL (Leica Biosystems, Nussloch, Germany model no. ST5010). Collagen types I and III, the most abundant protein found in skin, were evaluated using Picrosirius Red staining.

##### Picrosirius Red staining

Briefly, paraffin slides were de-waxed in two changes of xylene (Ajax Finechem; Thermo Fisher Scientific, cat. no. 576/2.5L/P) for three min, followed by one change of 100% ethanol for two min, 90%, 70% ethanol and distilled water for one min each. Slides were then incubated in Picrosirius Red (Picrosirius Red Stain Kit, Abcan, Melbourne, Australia, ab150681) for 60 min. Slides were then quickly rinsed in two changes of 0.5% acetic acid. Next, slides were hydrated in two changes of 100% ethanol for one min each, cleared in xylene and coverslipped with mounting media (Eukitt quick hardening mounting medium, Sigma-Aldrich (Merck), Melbourne, Australia, cat. no. 03989). After drying overnight, slides were then scanned at 20× using a 3DHistech Scan II Brightfield slide scanner (model Pannoramic Scan II, 3DHistech, Budapest, Hungary). Scanned images were exported using the Case viewer 2.2 platform. To evaluate differences between collagen type I and type III, fibres slides were analysed using polarised light with a Nikon Eclipse Ci microscope (Nikon, Tokyo, Japan) at 32.05 ms.

#### 2.5.2. Immunohistochemistry

Briefly, slides were deparaffinised in two changes of xylene for three min each, followed by one change in 100% ethanol for two min, and one change in 90%, 70%, 50% and Dako wash buffer (1:10, Dako (Agilent), Melbourne, Australia, cat. no. DM831) for one min each. Tissue samples were then delineated using a Dako pen (Dako (Agilent), Melbourne, Australia, cat. No. S2002). Enzyme-mediated antigen retrieval was performed in (i) 10 mM Tri-sodium Citrate buffer pH 6.0 (1.45 g Tri-sodium Citrate Dehydrate in 500 mL distilled water; pH to 6.0; add 250 μL Tween 20) or (ii) Tris-EDTA Buffer (10 mM Tris Base, 1 mM EDTA Solution, 0.05% Tween-20, pH 9.0, i.e., to make one litre: 1.21 g TRIS > add 0.37 g EDTA > add 1 L H2O > pH to 9.0 > add 0.5 mL Tween 20) heat immediate for five min at 95 °C; or using proteinase K (Dako (Agilent), Melbourne, Australia, cat. no. S3020) for five min. Next, 3% (*v*/*v*) hydrogen peroxidase was used to block endogenous peroxidase activity for five min. Slides were then washed in three changes of Dako wash buffer for two min each. Then, 2% (*w*/*v*) bovine serum (BSA, Sigma-Aldrich (Merck), Melbourne, Victoria cat. no. A7906) or sniper (Background sniper, Biocare Medical LLC, Redcliffe, Australia, cat. no. BS966) was used to block non-specific tissue protein binding for 30 min or 10 min, respectively. Slides were then incubated in primary antibody for 60 min. Next, slides were washed in three changes of Dako wash buffer for two min each and incubated in secondary antibody (EnVision+ dual-link system (Dako (Agilent), Melbourne, Australia, HRP rabbit/mouse kit, cat. no. K4061)) for 30 min. Slides were washed in Dako wash buffer three times for two min each wash. The immunological activity was detected via colour development with liquid diaminobenzidine (DAB) chromogen (Liquid DAB + substrate chromogen system, DAKO (Agilent), Melbourne, Australia, cat. no. K3468)). Slides were then washed in Dako wash buffer for one min and counterstained and coverslipped using a Leica Autostainer XL.

Prior to primary antibody incubation, the primary antibodies were diluted in 2% BSA. The following antibodies were used:

Collagen type I (COL I, 1:100, Abcam, Melbourne, Australia, cat. no. ab138492, RRID: AB_2861258) to evaluate collagen deposition. Antigen retrieval: 10 mM Tri-sodium Citrate buffer pH 6.0; DAB: 15 s.

von Willebrand Factor (vWF) (vWF, ready to use, Dako (Agilent), Melbourne, Australia, cat. no. IR52761-2, RRID: AB_2810304) to evaluate blood vessel formation. Samples were blocked with background sniper for 10 min, or BSA for 30 min. Antigen retrieval: proteinase K five min; DAB: two min and 30 s.

Alpha-smooth muscle actin (α-SMA, 1:1000, Abcam, Melbourne, Australia, cat. no. ab7817, RRID: AB_262054) to evaluate fibrous tissue encapsulation. Antigen retrieval: 10 mM Tri-sodium Citrate buffer pH 6.0; DAB: 15 s.

Cluster of Differentiation 68 (CD68, 1:300, Abcam, Melbourne, Australia, cat. no. ab125212, RRID: AB_10975465) to evaluate overall macrophagic activity. Samples were blocked with background sniper for 10 min. Antigen retrieval: Proteinase K five min; DAB: one min.

Inducible nitric oxide synthase (iNOS, 1:100, Abcam, Melbourne, Australia, cat. no. ab15323, RRID: AB_301857) to assess pro-inflammatory macrophage (M1) activity. Antigen retrieval: Tris-EDTA Buffer for vaginal tissue and 10 mM Tri-sodium Citrate buffer pH 6.0 for abdominal tissue; DAB: five min.

Mannose receptor (CD206, 1:100, Abcam, Melbourne, Australia, cat. no. ab8918, RRID: AB_306861) to evaluate pro-regenerative macrophage (M2) activity. Antigen retrieval was not performed. DAB: 20 s.

### 2.6. Scanning Electron Microscopy (SEM)

Scanning electron micrographs of the mPCL and PP meshes were acquired using a TM3000 Hitachi (Hitachi, Tokyo, Japan), at an accelerating voltage of 15.0 kV and a working distance of 5.2–5.6 mm.

### 2.7. Statistical Analysis

Quantitative assessment of the extent of collagen type I, vWF, CD68, iNOS and Mannose receptor immunostaining was performed using a MATLAB algorithm according to [35]. For the three-month time point, four samples (n = 4 images), and for the six-month time point, two samples (n = 2 images) were quantified. Five regions of interest (ROI) were quantified for each image sample group. This made for a total of 20 images for the three-month time point, and 10 for the six-month time point. An average of the five quantified stained areas was acquired, and analysis was performed using Microsoft Excel.

ROI size of 1000 × 1000 pixels was used to quantify collagen type I; ROI size of 2000 × 2000 pixels was used to quantify vWF and ROI size of 500 × 500 pixels was used to quantify CD68. ROI size of 1000 × 1000 pixels was used to quantify iNOS and Mannose receptors.

Statistical analysis was performed in SPSS v26 (IBM, Armonk, NY, USA). An independent *t*-test was performed comparing groups with normality assessed through Shapiro–Wilk test, skewness, and kurtosis. Log transformation was applied where specified. Data are re-ported as means ± standard deviations or standard errors of the mean. The significance level was defined as *p* < 0.05.

Due to regenerated tissue integration with the host over time, the new tissue formed was extremely well integrated, making delineation of the scaffold boundaries with native tissue challenge to distinguish, given the absence of a surrounding fibrous capsule. As such, to avoid inconsistency while quantifying fibrous tissue encapsulation via α-SMA re-activity, we have not quantified αSMA staining; only qualitative analysis was performed.

## 3. Results

### 3.1. Surgical Procedure & SEM

All animals recovered from general anaesthesia and surgical interventions without exhibiting any postoperative complications and completed the experimental period uneventfully.

It is known that due to the unique structural organisation of collagen fibres with a wavy pattern, soft tissues are characterised by very low stiffness and high flexibility when tested at low strain levels [36]. However, their stiffness increases significantly with increasing strains as the curvy collagen fibres become taut and they start carrying the applied loads very effectively. Additionally, soft tissues often exhibit a level of structural and mechanical anisotropy as their collagen fibres are arranged in one prominent direction. Inspired by the collagen architecture of native soft tissues, we designed scaffolds featuring both curvy microfibers and straight fibres. As shown in the scanning electron micrographs (Figure 1A,B), MEW technology enabled the realisation of this innovative mesh design which provides both flexibility and mechanical support which are needed for pelvic organ prolapse repair. We also added reinforcing fibres around the meshes to improve their suture retention and overall biomechanical properties of the meshes. SEM images indicated that the intended fibre architecture consisting of both straight and curvy fibres was achieved in the mPCL meshes. Fibre stacking was generally very accurate and only a small number of fibres were identified between the pores. The spacing and alignment of the fibres were found to be regular throughout the meshes. The pore size of mPCL meshes (500 μm × 1000 μm) was designed to be smaller than that of PP meshes (2500 μm × 2500 μm) as shown in Figure 1A and Figure 1C, respectively. The fibre diameter of the mPCL meshes was measured to be 43.6 ± 7.7 μm (Figure 1B), which is substantially smaller than that of PP meshes (142.0 ± 3 μm) (Figure 1D). The smaller fibre diameter leads to structures with higher surface area-to-volume ratio and is expected to enhance the integration of the implants with the body.

### 3.2. Mechanical Testing

The mechanical testing results (Figure 2) performed on the mesh–tissue complexes at the end of the in vivo study showed that mPCL mesh provides a good reinforcement effect to the implantation area. In contrast to the PP mesh, which is inherently very strong, our initial findings indicated that the new tissue formed within the pores of the mPCL mesh acts as a natural support for the weakened soft tissue. This is expected to reduce the long-term biocompatibility issues associated with conventional permanent meshes. Both PP and mPCL meshes exhibited a non-linear deformation behaviour with a J-shaped force-displacement curve where increasing force values were observed with increasing levels of displacements. This closely resembles the deformation behaviour of native soft tissues. Overall, PP meshes were found to be stiffer than mPCL scaffolds meshes both at the toe and linear regions. Both mPCL mesh/tissue and PP mesh/tissue complexes were thicker than their control counterparts, which contributed to their higher stiffness values. The findings were consistent for both implantation areas. As shown in our previous studies, our manufacturing technology is able to produce meshes with significantly different designs and mechanical properties, where decreases in the pore size and increases in the fibre diameter led to higher stiffness values [33]. The stretchability of the meshes can be enhanced by increasing the degree of curvature. The influence of architecture (pore size, degree of curvature, fibre diameter, etc.) on the mechanical properties of the resulting meshes can be found in [33].

### 3.3. Histological, Immunohistochemical and Histomorphometric Analysis

To harvest the implanted meshes aligned with the native orientation of the tissues, non-resorbable tacking sutures were placed adjacent to the implanted meshes as orientation markers for post-euthanasia specimen collection. The time points were chosen to fully evaluate temporal effects of host–biomaterial interactions, tissue repair, and healing. As chronic inflammation has been associated with delayed tissue healing and failure of scaffold integration, the three month time point was chosen to assess the early stages of foreign body reaction and tissue formation upon mesh implantation, whereas the six-month time point was chosen to evaluate the late-stage immune responses, as well as, outcomes of tissue repairing, architecture, and remodelling.

#### 3.3.1. Vaginal Tissue Site

H&E (Figure 3A–D), Masson trichrome staining (Figure 3E–H), and Picrosirius Red (Figure 3I–L) of the vaginal walls showed that the PP mesh threads and mPCL scaffold struts (red dashed line) were well integrated within the new fibrous tissue formed. Polarised light microscopy of the PP mesh group showed orderly, thicker fibres densely cluttered around the mesh threads (Figure 3I,K, white arrows), whereas in the mPCL group, the fibres seemed to be thinner and predominantly arranged within and around the scaffold struts (Figure 3J,L). Overall, the extent of the cell infiltration around the PP and mPCL meshes at three and six months post-mesh implantation did not differ.

Although significant temporal differences in collagen content were found for the mPCL group at six months, no differences were found for collagen type I deposition between the mPCL and PP groups (Figure 4A–D and Figure 5A).

Immunoreactivity of the α-SMA stain of the PP and mPCL groups was similar at three- and six-month time points, (Figure 4E,G and Figure 4F,H, respectively), with strong reactivity at vascular smooth muscle cells (blood vessels, black arrows) and at some myofibroblasts surrounding the meshes (red arrows). A primary round shape and sprouting nature of the newly formed blood vessels was observed at three months (Figure 4I,K, red arrows), and they appeared to be elongated and concentrically remodelled around the PP mesh threads (red dashed lines), as well as aligned to the collagen fibers (green dashed lines) after six months (Figure 4K). Yet, the round-shaped and evenly spread capillaries formed around the mPCL struts did not change over time (Figure 4J,L). Although there were no significant differences in neovascularisation between the mesh groups, the amount of blood vessel formation was found to be significantly higher in the mPCL group at six months (Figure 5B).

Macrophage infiltration, especially of the ones detected via CD68 reactivity, was mostly visible within the loose connective tissue around the PP and mPCL meshes, and in the cells lining the outer surface of the PP threads and mPCL struts at three months (Figure 4M and Figure 4N, respectively). After six months, the macrophage infiltration appeared to be localised at the cells lining the outer surface of the PP mesh threads, as opposed to the continuous expression at the same areas previously observed for the mPCL group (Figure 4O and Figure 4P, respectively). Although the amount of CD68 reactivity appeared to be inversely proportional between groups, yet higher for the PP mesh group, no significant differences were found over time (Figure 5C). iNOS (M1) reactivity was primarily observed at the mononucleated cells at three months, and at multinucleated cells within the soft tissue at later time point for the PP mesh group (Figure 4Q,R, red arrows). Although, iNOS expression was mostly neglected at three months for the mPCL group, at six months, reactivity was mainly observed at the epithelial cells lining blood vessels (Figure 4S,T, green arrows). iNOS expression increased over time in both groups, with a significant increase for the PP mesh group (Figure 4Q–T and Figure 5D). The immunoreactivity of the mannose receptor (M2) also appears to increase over time and to be yet localised to the cells lining the outer surface of the PP threads and mPCL struts (Figure 4U–X). Although the mannose receptor expression increased over time in both mesh groups, no significant differences were found (Figure 5E). The M1:M2 ratio for the mPCL and PP groups was higher at six months, 1.5:1 and 2.2:1, when compared to three months, 0.07:1 and 0.33:1, respectively (Figure 5F).

#### 3.3.2. Abdominal Tissue Site

No substantial physiological foreign body reaction was observed through H&E for the PP mesh and mPCL mesh at any time point for the abdominal wall; however, remnants of the PP threads and mPCL struts could be clearly recognised (Figure 6A–D, red dashed lines). Masson trichrome staining of the PP mesh group showed dense regular collagen fibers arranged linearly, mostly around the PP threads (red dashed lines), especially at six months (Figure 6E,G). The mPCL mesh group showed a less dense irregular arrangement of collagen fibers around the mPCL struts (red dashed lines) at three months; however, at six months, the fibers displayed a denser collagen fiber organisation and higher cellular infiltrate within the new tissue formed (Figure 6F and Figure 6H, respectively).

The polarised light microscopy (Figure 6I–L) of the abdominal wall of the three- and six-month time points also revealed temporal morphological differences in collagen formation for the PP and mPCL groups. New and thinner collagen fibers (immature collagen), depicted by the green/yellow colour, were observed for the mPCL group at three months (Figure 6J), whereas at six months, increased collagen type I fibers, (mature collagen) depicted by the orange and red stain colours (Figure 6L) were predominant. Conversely, the new tissue formation found in the PP mesh group appeared to be mature with substantially denser collagen fibers aligned in parallel to the implanted PP material (white dashed lines), especially at the six-month time point group (Figure 6K, white arrows).

In line with the histological staining results, collagen type I content was initially higher for the PP mesh group as opposed to the mPCL group. Although no significant differences were observed between groups, larger amounts of Collagen Type I deposition were observed for the mPCL group after six months (Figure 7A–D and Figure 8A).

Intense immunoreactivity of α-SMA was detected at the collagen fibers around the PP mesh threads at three- months, and to a less extend around the mPCL mesh (Figure 7E and Figure 7F, respectively). Vascular smooth muscle cells within the same areas were also positively stained at blood vessels (Black arrows). This immunoreactivity at the collagen fibers appeared to decrease over time for both groups (Figure 7G,H); however, smooth muscle cells were still intensely labelled at blood vessels, and at some myofibroblasts (red arrows) around the PP mesh threads. These newly formed blood vessel networks were positively stained by vWF (Figure 7I–L, red arrows), and seemed to be radially distributed around the PP threads and mPCL struts and parallel to the collagen fibers orientation (Figure 7I–L, green dashed line). The blood vessel density decreased for both groups after six months and was found to be significantly lower for the mPCL group (Figure 8B), potentially owing to the gradual vascular remodelling from capillary sprouting at three months, to a more mature anastomosed and elongated vessels over time (Figure 7I–L).

No signs of acute inflammation in the PP and mPCL treatment groups after three- and six months post meshes implantation were detected through CD68 staining; however, the presence of a few giant cells was detected around the PP mesh threads and around the mPCL struts at three months (Figure 7M,N, red arrows). Furthermore, it was observed that macrophages were located at the cells lining the outer surface and within the loose connective tissue surrounding the PP mesh, in contrast to the mPCL mesh, where they were primarily located around the mPCL struts at three months (Figure 7M–P). At six months, this expression appeared to increase within the fibrous tissue around the PP mesh threads and mPCL struts (Figure 7O,P). Although CD68 immunoreactivity was higher at six months in both groups, no significant differences were found (Figure 7M–P and Figure 8C).

Immunoreactivity of iNOS (M1) was detected only at mononucleated cells around and within the fibrous tissue surrounding the PP mesh threads at three months (Figure 7Q), whereas the reactivity of iNOS for the mPCL mesh group was mainly observed at fibroblasts within the fibrous tissue surrounding the mPCL struts (Figure 7R). At six months the immunoreactivity of iNOS significantly decreased for the PP group, whilst for the mPCL group, the immunoreactivity of iNOS was neglected (Figure 7S,T and Figure 8D). Mannose receptor (M2) reactivity appeared to be more prominent within the cell infiltration at three months, particularly at some giant cells located at the outer surface of the PP mesh threads and mPCL struts (Figure 7U and Figure 7V, respectively). This expression decreased over time for both groups and appeared to be predominantly observable at the outer surface of the PP mesh threads and mPCL struts (Figure 7W,X and Figure 8E). The extent of M1:M2 macrophage ratio was lower for both groups (PP: 0.1:1 at three months, 0:1 at six months; mPCL: 0.04:1 at 3 months, 0:1 at six months), with significant differences observed for the mPCL group at three and six months (Figure 8F).

## 4. Discussion

Polypropylene meshes have been recognised for their ability to induce the formation of chronic fibrosis, leading to the mesh becoming bridged and encapsulated as a result of its biologically inert nature. This process ultimately results in the development of a rigid scar and decreased flexibility of the newly formed tissue. In a previous study conducted by Hympánová et al. [37], it was suggested that non-degradable PP meshes possess inherent inflexible mechanical properties that may not be compatible with host tissues. In the context of scaffold-guided tissue engineering, achieving functional tissue restoration requires a harmonious match between the mechanical properties of the scaffolds and host tissues. Additionally, mechanical anisotropy plays a critical role in the success of the mesh implant. By incorporating anisotropic properties into the mesh design, such as variations in fiber orientation or architecture, the mesh can better emulate native tissue characteristics and ultimately enhance the overall success of the implantation process. To achieve anisotropic mechanical properties, the design of mPCL meshes incorporated both curvy and straight fiber architectures. Curvy fibres promote flexibility and straight fibres provide firmness, eventually leading to better mechanical conformity with the underlying native tissue.

In a study comparing lightweight and heavyweight meshes in rabbits, Bellón et al. [38] observed that mechanical properties were influenced by tissue site specificity and that collagen deposition depended on mesh pore size. Similarly, our study also revealed that the distinctive architecture of the PP and mPCL meshes, including variation in the pore size, substantially played a role in collagen formation and subsequent integration with the surrounding abdominal and vaginal tissue sites. The smaller pores and closer distances between mPCL mesh inter-struts, resulting in a higher surface area to volume ratio, facilitated collagen deposition, and enhanced early mesh/tissue integration. This, in turn, synergistically provided greater mechanical stability for constructive tissue remodelling at later time points. In contrast, the large PP mesh pores required rapid cell colonisation, which, in turn, prompted accelerated collagen synthesis. This increased collagen deposition leads to greater stiffness, as evident in the biomechanical results observed for the PP mesh group and explaining the observed decrease in mechanical properties of the abdominal wall over time for the mPCL group. The use of PCL fibers may have provided support for mesh and fascia fusion without causing a substantial increase in local stiffness, as observed by Plencner et al. [28]. In contrast, the mechanical properties of the mPCL mesh at the vaginal site exhibited a slight increase over time. This finding aligns with the study conducted by Diedrich et al. [39] where they also observed enhanced stiffness for a fully absorbable poly-4-hydroxybutyrate compared to a PP mesh in their evaluation of repairing vaginal prolapse in a sheep model after six months. The authors attributed the greater stiffness to tissue site-specificity and degradation processes of the poly-4-hydroxybutyrate scaffold, which has a similar degration profile to that of mPCL scaffolds.

The quality of extracellular matrix (ECM) deposition relies on various factors, including fibronectin and collagen content. These factors, in turn, also affect the mechanical properties of the tissue and are particularly influenced by tissue site specificity and mesh material. Gabriel et al. [40] conducted a study revealing significant biomechanical differences between abdominal and vaginal tissues, with the abdominal wall exhibiting four to ten times greater strength compared to the vaginal wall tissue. As observed with mechanical property outcomes, our histological and immunohistochemical analyses also revealed distinct tissue responses to mesh architecture and material between abdominal and vaginal tissue sites. In the PP mesh group, collagen content decreased in both abdominal and vaginal sites after six months. However, in the mPCL group, collagen deposition increased over time in the abdominal site while displaying a significant decrease in the vaginal site. When analysed based on collagen fiber orientation, the newly formed abdominal tissue in the mPCL group exhibited a dense irregular arrangement of collagen fibers, a common type of arrangement seen in supportive tissue of the skin. In contrast, the tissue formed around and within the PP mesh threads initially appeared loose but was later replaced to bundled fibrillar collagen fibers that were aligned and tightly packed parallel to the PP mesh threads. These features are crucial for achieving maximal tensile strength but also indicate the encapsulation and separation of the mesh by host tissue. Although, fibrosis is mainly characterised by progressive collagen deposition, which leads to hardening and tissue scaring [41], the smaller pores of the mPCL mesh enhanced cell attachment, while the larger pores supported tissue formation and orientation without bridging fibrotic tissues.

Furthermore, distinct temporal responses using the mPCL mesh scaffold could be observed for the vaginal group. At three months, the mPCL group promoted the formation of new and thin collagen fibers whereas at six months, mature and thicker collagen fibers were predominant. Increased synthesis of ECM and collagen formation promotes faster collagen deposition on the surface of the mesh, resulting in poor tissue integration, and decreased mechanical properties [42,43]. Although increased collagen deposition could augment tensile strength and mechanical stiffness of the repaired abdominal and vaginal walls, excessive collagen synthesis could also lead to decreased elasticity of the regenerated tissue [44]. Initial collagen type I followed by collagen type III deposition is desired, so that immature collagen is remodelled throughout the three-dimensional collagen network, thus promoting superior biomechanical properties to the newly built tissue.

Key immunomodulatory aspects of tissue repairing include the regulation of inflammation, foreign body reaction, and homeostasis. While macrophages are primarily recognised for their phagocytic capacity, they also engage with myeloid, endothelial, pericytes and vascular smooth cells to promote wound healing and tissue repair [45]. In the newly formed abdominal tissue, the CD68 monocytic profile was retained, and higher levels of M2 (pro-regenerative) macrophagic polarisation was observed over time for both groups, suggesting an introductory inflammatory reaction followed by early stages of tissue repairing. Contrary to the abdominal wall, M1 was higher than the M2 macrophagic polarisation at six months for the vaginal wall. It has been suggested that the M2 macrophagic polarisation is essential for cell differentiation into several phenotypes with function in tissue deposition and remodelling, while M1 macrophagic polarisation is critical for angiogenesis. Depletion of macrophages, especially M1, can hinder neo-vascularisation and lead to scarring in later stages of tissue repair.

In contrast to acute inflammatory processes which require constant vascular changes to promote infiltration of cells, particularly macrophages, fibrosis usually originates from established inflammation processes and is characterised by the presence of active fibroblasts positive for α-smooth muscle actin (α-SMA). ECM synthesis and turnover are largely attributed to the activity of these contractile cells. It has been suggested that M1 macrophages, besides exacerbating inflammatory reactions, also contribute to the proliferation of active fibroblasts positive for α-SMA [46,47]. Our results corroborate previous research, further indicating a correlation between the expression of M1 and α-SMA in the abdominal and vaginal walls. However, in contrast to the existing literature [48], our study did not observe fibrotic encapsulation based on the expression of α-SMA. Instead, we noticed that the macrophages lining the outer surface of the mPCL struts were attached to mesh rather than forming a fibrin matrix around it. It could be speculated that these cells play a role in the degradation of the PP and mPCL mesh remnants, potentially reversing fibrosis formation [41]. Interestingly, our findings further indicate that both M1 and M2 co-existed during at both the three- and the six-month time points, challenging the widely advocated M1–M2 distinction. During the early stages of healing, it is crucial to retain the mesh implant for mechanical support. However, the continued presence of mesh threads may trigger a robust foreign body reaction, thus explaining the higher levels of M1 macrophages observed for the PP mesh group at vaginal site [44].

The arrangement of collagen fibers in both the mPCL and PP meshes was also found to influence the organisation of the neovascular network over time. Specifically, blood vessels were predominantly observed around the PP mesh, while neo-vessels formed within the mPCL fibers. The presence of densely packed collagen fibers around the PP mesh appears to guide the formation and remodelling of larger and elongated vessels around the PP mesh, thus explaining the slightly higher levels of vWF observed in the abdominal and vaginal walls at the six-month mark when compared to mPCL group. These findings suggest that the variations in mesh architecture contribute to these results, highlighting the influence of mesh structure on tissue response and vascularisation.

Restoration of functional normal tissue architecture requires an early transient influx of monocytes from the host bloodstream followed by macrophage differentiation to activate angiogenesis processes to form a new vasculature. Emerging evidence demonstrates that new tissue formation also requires substantial vascular remodelling [41]. Corliss et al. [49] and Spiller et al. [50] have highlighted the importance of macrophages to angiogenesis processes, including anastomoses of sprouting blood vessels. This highly coordinated process is crucial to support cell recruitment, proliferation, differentiation, and collagen deposition [51], in order to create a framework strong enough to support vascular remodelling, new tissue formation and to sustain tensile forces. Additionally, it has been suggested that mesh orientation and load transfers along the mesh architecture direct tissue formation and organisation [52].

mPCL meshes are specifically designed to guide tissue formation and eventually be replaced by functional tissue in the long term. The use of a slow-degrading material like mPCL may overcome the drawbacks associated with PP meshes, resorbable coatings, and xenografts, including loss of strength and fibrosis formation.

Preliminary results, at the end of this pilot study after six months, suggest that the surgical implantation of the PP and mPCL meshes is safe with no morbidity, including no signs of wound infection or graft erosion into surrounding viscera. As the pilot results are supportive of mPCL as a safe and effective biodegradable scaffold for hernia and vaginal prolapse repair, a full-scale long-term study over a 24–36 month period with an adequate sample size is recommended.

## 5. Conclusions

This pilot study aimed to test the safety and effectiveness of mPCL-based biodegradable scaffolds for hernia and vaginal prolapse repair. As this concept primarily relies on the body to build new tissue and create natural reinforcement in herniated tissues, PRP was used to enhance the bio-functionality of our inert mPCL scaffolds, which are known to be biologically active. Future studies will focus on exploring the effects of PRP as well as the design features (pore size, fibre diameter, thickness etc.) of the mPCL scaffolds in vivo.

## Figures and Tables

**Figure 1 bioengineering-10-01242-f001:**
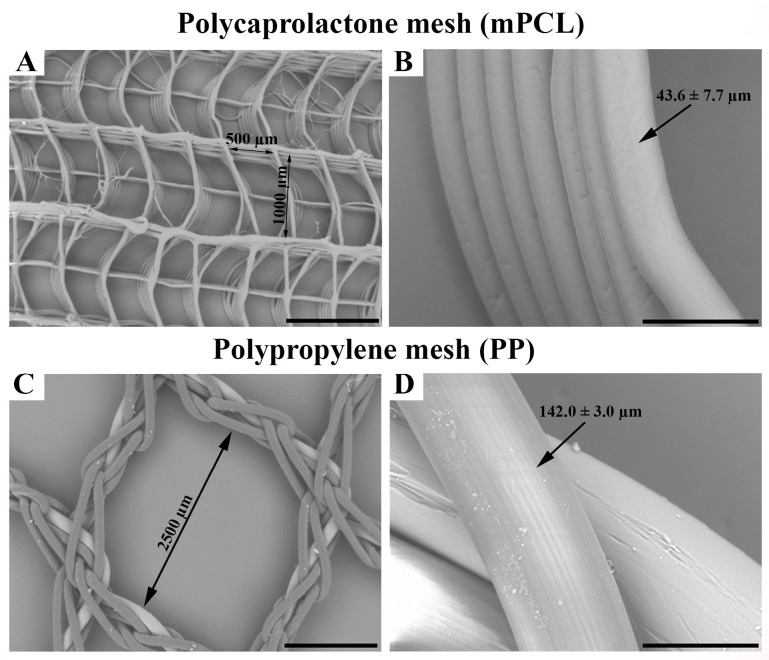
Scanning electron microscopic (SEM) images showing the ultrastructural morphology of the medical grade polycaprolactone (mPCL) (**A**,**B**) and as well as polypropylene (PP) mesh scaffolds (**C**,**D**). Scale bars: (**A**,**C**) 1 mm; (**B**,**D**) 100 μm.

**Figure 2 bioengineering-10-01242-f002:**
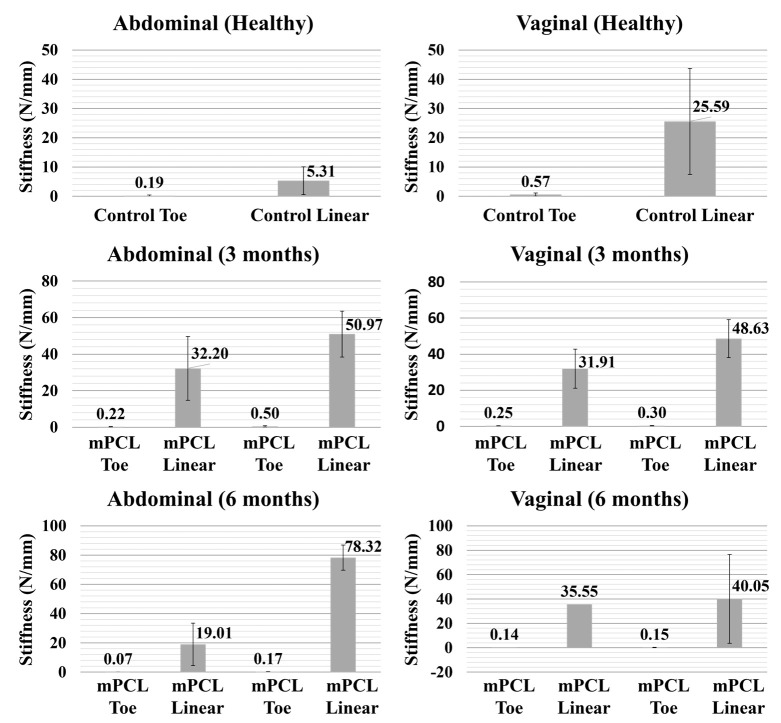
Stiffness of healthy abdominal and vaginal tissue (control), mPCL mesh/tissue complex and polypropylene mesh/tissue complex explanted from the abdominal and vaginal regions at different time points (3 and 6 months). The slope of the force–displacement curves was used to determine the stiffness of the tested specimens (the region between the displacement of 1 mm and 5 mm for the toe region; the region between the displacement of 7.5 mm and 15 mm for the linear region).

**Figure 3 bioengineering-10-01242-f003:**
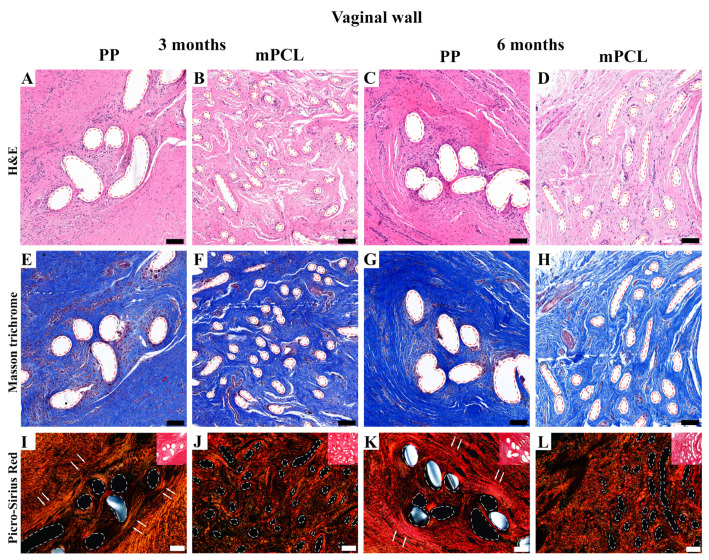
Histology overview of the vaginal wall implanted with the PP mesh and mPCL mesh scaffold at three- and six-month time points. (**A**–**D**) H&E; (**E**–**H**) Masson trichrome; (**I**–**L**) polarised light microscopy of Picrosirius Red staining (inset images) of the new tissue formed within the PP and mPCL meshes. Red and white dashed lines: PP mesh threads and mPCL struts. White arrows indicate collagen fibers alignment around PP nesh threads. Scale bars: 100 μm.

**Figure 4 bioengineering-10-01242-f004:**
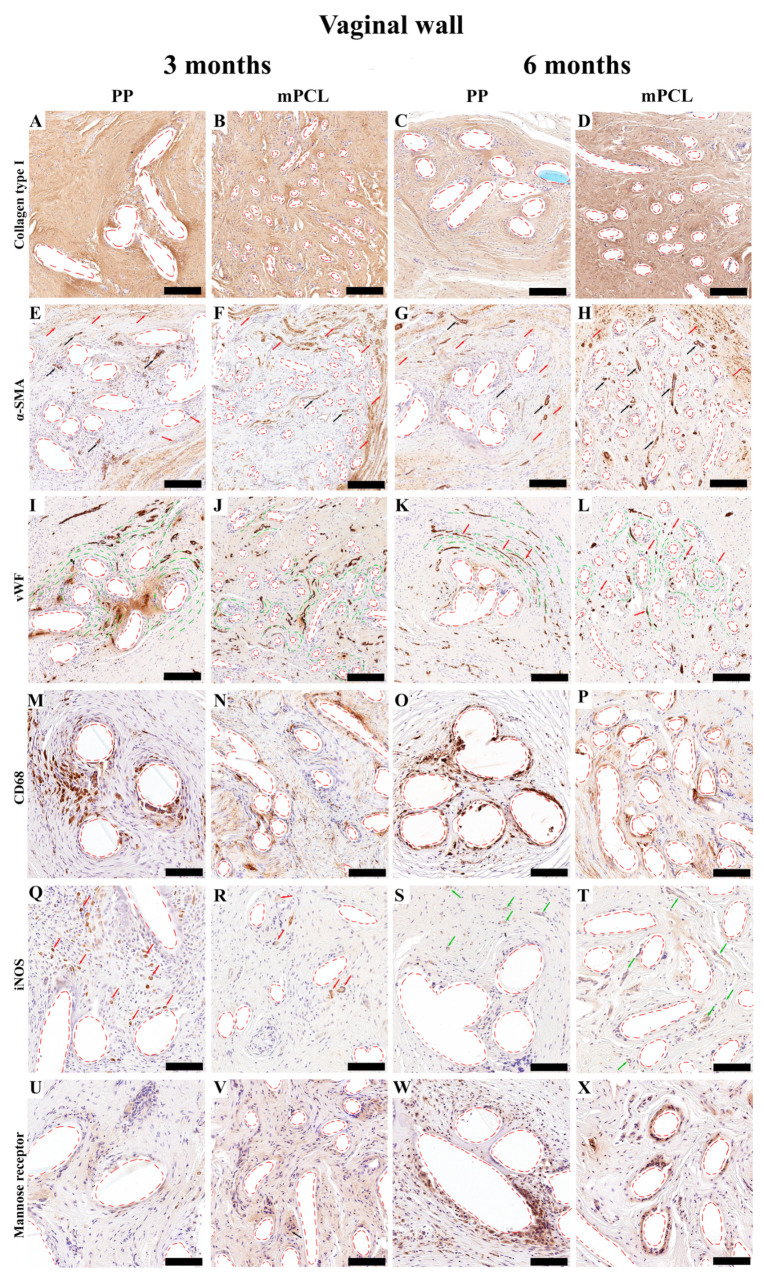
Immunohistochemistry overview of the vaginal wall implanted with the PP mesh and mPCL mesh scaffold at three- and six-month time points. The cellular responses of the new tissue formed was evaluated using collagen type I (**A**–**D**). Anti-smooth muscle actin (α-SMA); red arrows indicate fibroblast, and black arrows indicate blood vessels (**E**–**H**). von Willebrand factor (vWF); red arrows indicate blood vessels and green lines indicate collagen fibers alignment (**I**–**L**). Cluster of differentiation CD68 (CD68) macrophages, red arrows indicate macrophages and black arrows indicate fibroblast (**M**–**P**). Inducible nitric oxide synthase (iNOS) (M1); red arrows indicate monocytes and green arrows indicate blood vessels (**Q**–**T**). Mannose receptor (M2) (**U**–**X**). Red dashed lines: PP mesh threads and mPCL struts. Scale bars: (**A**–**L**) 200 μm; (**M**–**X**) 100 μm. Isotypes are provided in Appendix A.

**Figure 5 bioengineering-10-01242-f005:**
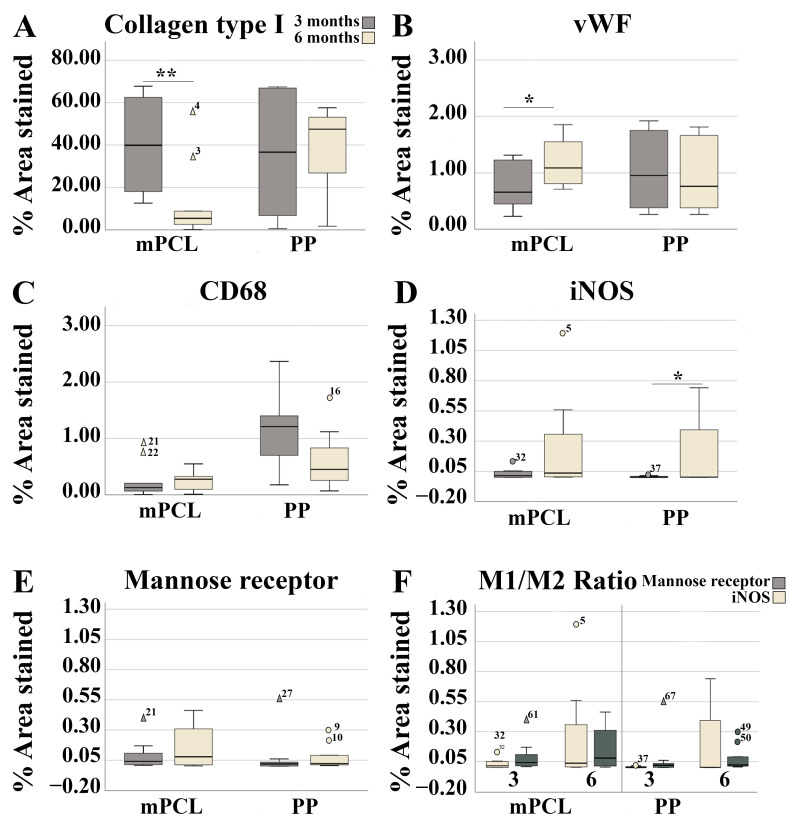
Histomorphometric evaluation of the vaginal wall implanted with the PP mesh and mPCL mesh scaffold at three- and six-month time points for (**A**) collagen type I, (**B**) vWF, (**C**) CD68, (**D**) iNOS, (**E**) Mannose receptor and (**F**) M1:M2 ratio. Data are reported as means ± standard deviations or standard errors of the mean. The umbered circles in the graph represent outliers, while numbered triangles are denoting extreme outliers. The numerical values associated with these symbols reflect the actual values of these data points within the dataset. The significance level was defined as *: *p* < 0.05, **: *p* < 0.01.

**Figure 6 bioengineering-10-01242-f006:**
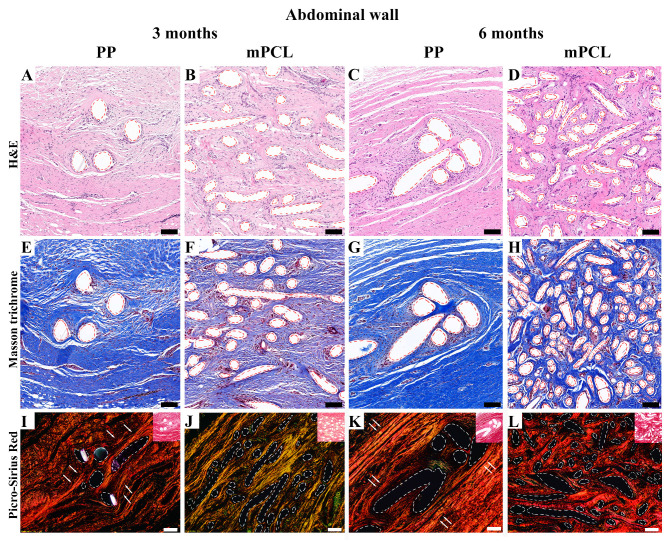
Histology overview of the abdominal wall implanted with the PP mesh and mPCL mesh scaffold at three- and six-month time points. (**A**–**D**) H&E; (**E**–**H**) Masson trichrome; (**I**–**L**) polarised light microscopy and Picrosirius Red staining (inset images): of the new tissue formed within the PP and mPCL meshes. Red and white dashed lines: PP mesh threads and mPCL struts. White arrows indicate collagen fibers alignment around PP nesh threads. Scale bars: 100 μm.

**Figure 7 bioengineering-10-01242-f007:**
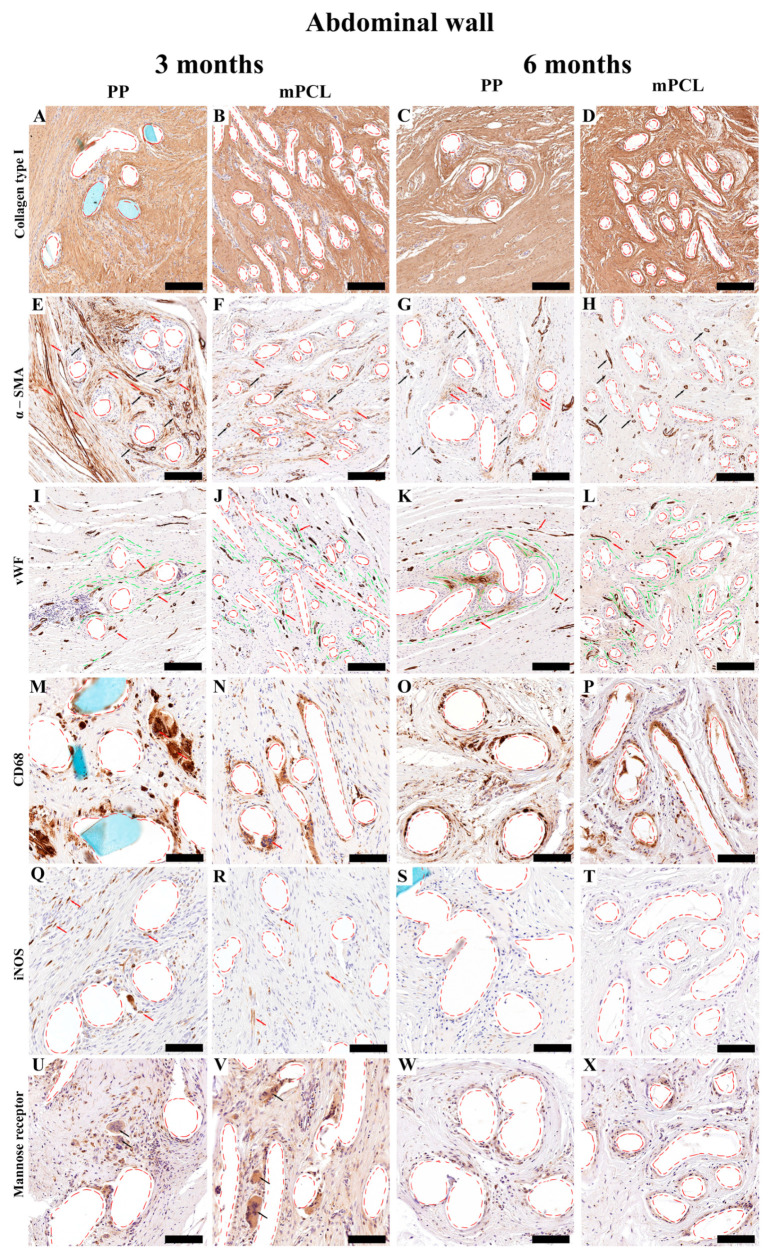
Immunohistochemistry overview of the abdominal wall implanted with the PP mesh and mPCL mesh scaffold at three- and six-month time points. The cellular responses of the new tissue formed was evaluated using collagen type I (**A**–**D**). Anti-smooth muscle actin (α-SMA); red arrows indicate fibroblast, and black arrows indicate blood vessels (**E**–**H**). von Willebrand factor (vWF); red arrows indicate blood vessels, and green lines indicate collagen fibers alignment (**I**–**L**). Cluster of differentiation 68 (CD68) macrophages; red arrows indicate giant cells (**M**–**P**). Inducible nitric oxide synthase (iNOS) (M1); red arrows indicate monocytes (**Q**–**T**). Mannose receptor (M2); black arrows indicate giant cells (**U**–**X**). Red dashed lines: PP mesh threads and mPCL struts. Scale bars: (**A**–**L**) 200 μm; (**M**–**X**) 100 μm. Isotypes are provided in Appendix A.

**Figure 8 bioengineering-10-01242-f008:**
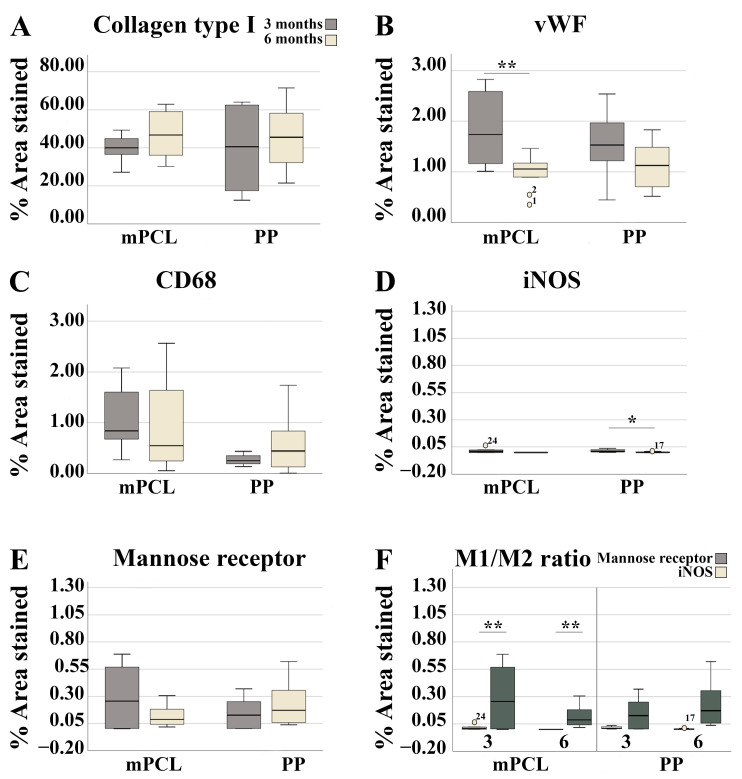
Histomorphometric evaluation of the abdominal wall implanted with the PP mesh and mPCL mesh scaffold at three- and six-month time points for (**A**) collagen type I, (**B**) vWF, (**C**) CD68, (**D**) iNOS, (**E**) Mannose receptor, and (**F**) M1:M2 ratio. Data are reported as means ± standard deviations or standard errors of the mean. The numbered circles in the graph represent outliers. The numerical values associated with these symbols reflect the actual values of these data points within the dataset. The significance level was defined as *: *p* < 0.05, **: *p* < 0.01.

## Data Availability

The data that support the findings of this study are available from the corresponding author upon reasonable request.

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
