# Peer review of "3D-Printed Medical-Grade Polycaprolactone (mPCL) Scaffold for the Surgical Treatment of Vaginal Prolapse and Abdominal Hernias"

_bioengineering, 2023, doi:10.3390/bioengineering10111242_

Round 1

Reviewer 1 Report

The provided paper presents a study involving the fabrication of mPCL mesh scaffolds, their preparation with platelet-rich plasma (PRP), surgical implantation in sheep, and subsequent biomechanical and histological analyses. Here are some questions that can help evaluate the publication:

-How is the MEW technology used to fabricate the mPCL mesh scaffolds? Are there any specific advantages of this technology over other approaches?

- Are the sterilization steps comprehensive and validated? Were the effects of sterilization on scaffold properties considered?

-Can you elaborate on how the mechanical anisotropy achieved through curvy and straight fibers contributes to improved mesh performance compared to traditional PP meshes?

-Were there any efforts made to characterize the chemical and mechanical properties of the fabricated mPCL mesh scaffolds before implantation, and if not, how does this impact the interpretation of the study's results?

-How did the choice of pore sizes (500 µm and 1000 µm) for each direction (horizontal and vertical) contribute to mimicking the biomechanical behavior of native soft tissue?

- Can you elaborate on the potential interplay between the different types of collagen fibers and their organization observed in the histological staining, and how this relates to the mechanical performance and tissue integration of the meshes?

-Were there any challenges or limitations encountered during the scaffold fabrication process, and how were they addressed?

- Were there any unexpected complications during the surgical procedure, and if so, how were they managed?

- Were there any challenges in performing the plunger tests, such as potential effects of sample irregularities, and how were these challenges managed?

- Can you provide insights into the comparison of stiffness values between the different mesh types and their implications for tissue repair outcomes?

Author Response

The authors would like to thank the reviewer for taking the time to review our manuscript. We believe we have addressed all reviewer’s queries and have produced a better version of the manuscript. Please find the detailed responses below.

Reviewer’s queries are in black colour font and authors responses are in red colour font. All changes made within the manuscript have been highlighted in yellow throughout the re-submitted manuscript file.

Reviewer 1.

The provided paper presents a study involving the fabrication of mPCL mesh scaffolds, their preparation with platelet-rich plasma (PRP), surgical implantation in sheep, and subsequent biomechanical and histological analyses. Here are some questions that can help evaluate the publication:

-How is the MEW technology used to fabricate the mPCL mesh scaffolds? Are there any specific advantages of this technology over other approaches?

We thank the reviewer for this comment. The details of the MEW process and how it works can be found in reference [40 doi:10.1002/adma.201103482]. As discussed in the manuscript, MEW technology allows for manufacturing pre-defined scaffold architectures from micro-fibres. The ability to control the geometry facilitates the fabrication of highly customized designs suited for each application. Also, in contrast to other extrusion-based 3D printing systems, such as fused deposition modelling, MEW yields very fine fibres with a diameter of <50 microns. This small fibre diameter leads to scaffolds with high surface area-to-volume ratio, which can help with the tissue integration. The smaller pores of the mPCL mesh and its inherently smaller distances between mesh inter-struts and higher surface area would prompt collagen deposition and simultaneously provide mechanical stability for the new tissue to be formed and remodelled, as opposed to the larger interstices between the PP mesh pores.

We have captioned some of the MEW advantages at the following sections of the manuscript:

Manuscript Results section, Page 6, lines 294-297: “…MEW technology enabled the realisation of this innovative mesh design which provides both flexibility and mechanical support which are needed for pelvic organ prolapse repair. We have also added reinforcing fibres around the meshes to improve their suture retention and overall biomechanical properties of the meshes”.

Manuscript Results section, page 7-8, lines 325-331: “As shown in our previous studies, our manufacturing technology is able to produce meshes with significantly different designs and mechanical properties, where decreases in the pore size and increases in the fibre diameter leads to higher stiffness values [41]. The stretchability of the meshes can be enhanced by increasing the degree of curvature. The influence of architecture (pore size, degree of curvature, fibre diameter etc.) on the mechanical properties of the resulting meshes can be found in reference [41]”.

- Are the sterilization steps comprehensive and validated? Were the effects of sterilization on scaffold properties considered?

We thank the reviewer for this comment. “The scaffolds were sterilized by immersing them into 80% ethanol for 5 min and then allowing the ethanol to evaporate in a biosafety cabinet overnight. The scaffolds were also exposed to UV light exposure for 20 min” (Material and methods section, Page 3, lines 124-126). Although it has been shown that plasma treatment can lead to polymer degradation and increased brittleness of polycaprolactone MEW scaffolds [https://doi.org/10.1002/marc.202100433], there is no data available on the effects of sterilisation on these scaffolds. Yet, we believe it is an important topic which needs to be further explored in the future studies. We apologise to the reviewer as we have not properly referenced supplemental material in the main text. We have shown some of the effects of sterilization on scaffold properties in the supplementary material section of the manuscript (Figure S1). If the post treatment for surface hydrolyzation and/or sterilization is not performed according to protocol the mechanical properties of the scaffold are reduced significantly. We have now included the following in the material and methods section page 3, lines 126-129: “If the post treatment for surface hydrolyzation and/or sterilization is not performed according to protocol the mechanical properties of the scaffold can be reduced significantly (Figure S1)”.

-Can you elaborate on how the mechanical anisotropy achieved through curvy and straight fibers contributes to improved mesh performance compared to traditional PP meshes?

We thank the reviewer for this comment. The abdominal and vaginal wall muscles exhibit distinctive non-linear behaviour and fulfill a wide spectrum of mechanical functions due to their anisotropic (different directions) characteristic. These functions encompass the regulation of abdominal and vaginal movement and loading distribution in both horizontal and vertical directions. To more faithfully replicate the native tissue characteristics, particularly these complex biomechanical behaviors, we employed an mPCL mesh scaffold with fibres of both curvy and straight architectures and varied pore sizes (500 µm and 1000 µm) in each direction (horizontal and vertical). The use of curvy fibres provided flexibility to one loading direction, whereas straight fibres provided firmness in the other loading direction, leading to mechanical anisotropy and better tissue compliance. In contrast, polypropylene (PP) meshes, owing to their inherent nonabsorbable and inflexible mechanical properties, primarily aim to maintain structural integrity and reinforce the weakened tissue site through mechanical tension rather than promoting tissue regeneration.

To enhance clarity, we have included passages from the manuscript where we have elaborated on these points:

Manuscript Results section, Page 7, lines 286-293: “It is known that due to the unique structural organisation of collagens fibres with a wavy pattern, soft tissues are characterized by very low stiffness and high flexibility when tested at low strain levels [44]. However, their stiffness increases significantly with increasing strains as the curvy collagen fibres become taut and they start carrying the applied loads very effectively. Also, soft tissues often exhibit a level of structural and mechanical anisotropy as their collagen fibres are arranged in one prominent direction. Inspired by the collagen architecture of native soft tissues, we designed scaffolds featuring both curvy microfibers and straight fibres”.

Manuscript Results section, Page 7, lines 313-321: “The mechanical testing results (Figure 2) performed on the mesh-tissue complexes at the end of the in vivo study showed that mPCL mesh provides a good reinforcement effect to the implantation area. In contrast to the PP mesh, which is inherently very strong, our initial findings indicated that the new tissue formed within the pores of the mPCL mesh acts as a natural support for the weakened soft tissue. This is expected to reduce the long-term biocompatibility issues associated with conventional permanent meshes. Both PP and mPCL meshes exhibited a non-linear deformation behaviour with a J-shaped force-displacement curve where increasing force values were observed with increasing levels of displacements. This closely resembles the deformation behaviour of native soft tissues”.

Manuscript Discussion section, Page 16, lines 509-515: “By incorporating anisotropic properties into the mesh design, such as variations in fiber orientation or architecture, the mesh can better emulate native tissue characteristics and ultimately enhance the overall success of the implantation process. To achieve anisotropic mechanical properties, the design of mPCL meshes incorporated both curvy and straight fiber architectures. Curvy fibres promote flexibility and straight fibres provide firmness, eventually leading to better mechanical conformity with the underlying native tissue”.

Manuscript Discussion section, Page 16-17, lines 521-527: “The smaller pores and closer distances between mPCL mesh inter-struts, resulting in a higher surface area to volume ratio, facilitated collagen deposition and enhanced early mesh/tissue integration. This, in turn, synergistically provided greater mechanical stability for constructive tissue remodeling at later time points. In contrast, the large PP mesh pores required rapid cell colonization, which, in turn, prompted accelerated collagen synthesis. This increased collagen deposition leads to greater stiffness…”.

-Were there any efforts made to characterize the chemical and mechanical properties of the fabricated mPCL mesh scaffolds before implantation, and if not, how does this impact the interpretation of the study's results?

We thank the reviewer for this comment. Chemical characterization of the mPCL has been previously done by our group in: Dynamics of in vitro polymer degradation of polycaprolactone-based scaffolds: accelerated versus simulated physiological conditions (doi: 10.1088/1748-6041/3/3/034108) and in Evaluation of Polycaprolactone Scaffold Degradation for 6 Months in Vitro and in Vivo (doi:10.1002/jbm.a.32052). Briefly, the polymer phase of our mPCL scaffold would provide plasticity, as well as a slow degradation and resorption kinetics profile by hydrolysis of the PCL ester linkages at physiological pH and temperature. This in turn would direct tissue ingrowth and organization via physical growth guidance at early phases of the tissue repair.

This pilot study aimed at testing the in vivo performance of meshes and understanding if they lead to a natural mechanical reinforcement while minimizing the risks associated with permanent meshes. Therefore, we conducted our mechanical tests after the explantation. We have referred readers to our previous studies investigating the mechanical properties of our meshes in detail in page 2, line 96 Reference 41 (doi:10.1021/acsami.7b08617).

-How did the choice of pore sizes (500 µm and 1000 µm) for each direction (horizontal and vertical) contribute to mimicking the biomechanical behavior of native soft tissue?

We thank the reviewer for this comment. Firstly, the pore size of mPCL meshes (500 µm * 1000 µm) was designed to be smaller than that of PP meshes (2500 µm * 2500 µm). As described in the discussion section Page 16-17, lines 521-527: “The smaller pores and closer distances between mPCL mesh inter-struts, resulting in a higher surface area to volume ratio, facilitated collagen deposition and enhanced early mesh/tissue integration. This, in turn, synergistically provided greater mechanical stability for constructive tissue remodeling at later time points. In contrast, the large PP mesh pores required rapid cell colonization, which, in turn, prompted accelerated collagen synthesis. This increased collagen deposition leads to greater stiffness…”, as well as mesh encapsulation and fibrosis formation. However, the smaller pores featured in the mPCL mesh scaffold would mitigate these drawbacks by promoting and enhancing cell attachment. Concurrently, the larger pores would support tissue formation and orientation without bridging fibrotic tissues.

- Can you elaborate on the potential interplay between the different types of collagen fibers and their organization observed in the histological staining, and how this relates to the mechanical performance and tissue integration of the meshes?

We thank you the reviewer for this comment. The quality of extracellular matrix (ECM) deposition relies on various factors, including collagen content. This, in turn, affect the mechanical properties of the tissue and is particularly influenced by tissue site specificity (abdominal or vaginal) and mesh material (Page 17, lines 538-541). As observed with mechanical property outcomes, our histological and immunohistochemical analyses revealed distinct tissue responses to mesh architecture and material between abdominal and vaginal tissue sites (Page 17, lines 543-546). When analysed by collagen fiber orientation, the newly formed abdominal tissue in the mPCL group exhibited a dense (Collagen type I - mature) irregular arrangement of collagen fibers, a common type of arrangement seen in supportive tissue of the skin. In contrast, the tissue formed around and within the PP mesh threads initially appeared loose (Collagen type III - immature) but was later replaced to bundled fibrillar collagen fibers (mature tissue) that were aligned and tightly packed parallel to the PP mesh threads (Page 17, lines 549-554). Initial collagen type I followed by collagen type III deposition is desired, so that immature collagen is remodelled throughout the three-dimensional collagen network, thus promoting constructive and superior biomechanical properties to the newly built tissue (Page 17, lines 568-570).

-Were there any challenges or limitations encountered during the scaffold fabrication process, and how were they addressed?

We thank the reviewer for this comment. MEW is a well-established technology and as long as the users do not push its limits in terms of printing resolution, speed and accuracy, aim at scaffolds with reasonable complexity and sizes, it yields scaffolds in a highly reproducible manner. In this study, we considered these limitations and focused on obtaining scaffold architectures suited for our application without compromising reproducibility. Overall, as demonstrated by SEM images, the intended fibre architecture of both straight and curvy fibres was mostly uniform in the mPCL meshes. Fibre stacking was generally very accurate and only a small number of fibres were identified between the pores. The spacing and alignment of the fibres were also found to be regular throughout the meshes (described in Page 6, lines 297-301).

- Were there any unexpected complications during the surgical procedure, and if so, how were they managed?

We thank the reviewer for asking about unexpected complications during the surgical procedure. “All animals recovered from general anaesthesia and surgical interventions without exhibiting any postoperative complications and completed the experimental period uneventfully”, (described in Page 6, lines: 283-285).

- Were there any challenges in performing the plunger tests, such as potential effects of sample irregularities, and how were these challenges managed?

This is an excellent point raised by the reviewer. Due to regenerated tissue integration with the host over time, the new tissue formed was extremely well integrated making delineation of the scaffold boundaries with native tissue challenge to distinguish, given the absence of a surrounding fibrous capsule. As such, “Some of the specimens had to be excluded due to the major irregularities in their shape, size and/or thickness” (described in Page 4, lines: 180-181 of the manuscript). High level of attention was paid during the surgical removal, preparation as well as fixation of the specimens to the mechanical tester to reduce irregularities. Also, we applied a tare load at the beginning of the mechanical tests as described below.  

“A tare load of 0.05 N was applied for determining the starting point of the test during the analysis of the tests. This tare load excluded minor force readings arising from the surface features of the specimens and ensured that full contact between the plunger and samples was achieved.” described in Page 4, lines: 174-177 of the manuscript).

- Can you provide insights into the comparison of stiffness values between the different mesh types and their implications for tissue repair outcomes?

We thank the reviewer for this comment. Mesh related complications are influenced by the mesh architecture and its mechanical properties. Gradual increase of stiffness has been linked to time dependent variations in biomechanical properties of PP meshes, likely due to incorporation of the mesh within the surrounding tissue overtime [doi:10.1097/01.ju.0000121377.61788.ad] (Page 2, lines 58-62). The large pore size of these meshes would induce cells to promptly synthesize collagen within the mesh threads, thus causing an increase in stiffness and decrease in mechanical strength, leading to tissue hardening and resulting in mesh contraction, exposure, and persistent pain [doi:10.1097/01.ju.0000060119.43064.f6, 10.1016/j.surg.2008.04.005, doi:10.1016/j.ajog.2006.07.006, doi:10.1016/j.ajog.2008.12.040, doi:10.1111/1471-0528.12081] (Page 2, 62-65). The large PP mesh pores required rapid cell colonization, which, in turn, prompted accelerated collagen synthesis. This increased collagen deposition leads to greater stiffness, as evident in the stiffness results observed for the PP mesh group and explaining the observed decrease in stiffness of the abdominal wall over time for the mPCL group. The use of PCL fibers may have provided support for the mesh scaffold and fascia fusion without causing a substantial increase in local stiffness as observed by Plencner et al. [doi:10.2147/IJN.S63095] (Page 17, 525-531). In contrast, the stiffness of the mPCL mesh at the vaginal site exhibited a slight increase over time. This finding aligns with the study conducted by Hympánová et al. [doi:10.1111/1471-0528.17040] where they also observed enhanced stiffness for a fully absorbable poly-4-hydroxybutyrate compared to a PP mesh in their evaluation of repairing vaginal prolapse in a sheep model after six months. The authors attributed the greater stiffness to tissue site-specificity and degradation processes of the poly-4-hydroxybutyrate scaffold, which has a similar degration profile to that of mPCL scaffolds (described in Page 15, lines 531-544) (Page 17, 531-537).

Reviewer 2 Report

The current manuscript described a comprehensive evaluation of the 3D printed polycaprolactone (mPCL) scaffold for 2 the surgical treatment of vaginal prolapse and abdominal hernias. The conclusion is sound and well supported by the data. I only have a few minor comments:

1, it will be great if the authors can provide a schematic design at the beginning to introduce their experimental design and methods.

2, In figure 1, why did the authors adopt the current dimension and parameters for the 3D printed scaffolds?

3, How many replicates were measured in figure 2? The same issue for figure 4 and 6.

4, In figure 4 and figure 6, they y axis value is not readable. The statistical analysis is also not readable. In addition, why there is no quantified results for alpha-SMA?

Author Response

The authors would like to thank the reviewer for taking the time to review our manuscript. We believe we have addressed all reviewer’s queries and have produced a better version of the manuscript. Please find the detailed responses below.

Reviewer’s queries are in black colour font and authors responses are in red colour font. All changes made within the manuscript have been highlighted in yellow throughout the re-submitted manuscript file.

Reviewer 2.

The current manuscript described a comprehensive evaluation of the 3D printed polycaprolactone (mPCL) scaffold for 2 the surgical treatment of vaginal prolapse and abdominal hernias. The conclusion is sound and well supported by the data. I only have a few minor comments:

1, it will be great if the authors can provide a schematic design at the beginning to introduce their experimental design and methods.

We thank the reviewer for the suggestion. We have included a schematic design in the supplementary material (Figure S1) section.

2, In figure 1, why did the authors adopt the current dimension and parameters for the 3D printed scaffolds?

We thank the reviewer for the interest. The meshes were designed to mimic the complex biomechanical behaviour of the native soft tissue, such as anisotropy and non-linearity. To achieve this, fibres with curvy and straight architectures were used in the meshes, and different pore sizes (500 µm and 1000 µm) were employed for each direction (horizontal and vertical). The use of curvy fibres provided flexibility to one loading direction, whereas straight fibres provided firmness in the other loading direction, leading to mechanical anisotropy. Moreover, as we responded to the comment of reviewer 1 above, we considered the limitations of the MEW technology and did not aim for manufacturing scaffolds with smaller geometrical features to ensure quality and reproducibility.

3, How many replicates were measured in figure 2? The same issue for figure 4 and 6.

We thank the reviewer for this comment. For image 2 we have quantified for the three months’ time point samples (n=3) as there was a specimen that was not suitable for mechanical testing due to the major irregularities in their shape, size and/or thickness) and for the six months’ time point samples (n=2). For figure 4 and 6, we have quantified for the three months’ time point samples (n=4) and for the six months’ time point samples (n=2) and five regions of interest per image. This totalizes for the three months’ time point 20 images and for the six months’ time point 10 images. We have amended and highlighted our description to make it clearer in page 6, lines 262-265 of the manuscript as follow: “For the three months’ time point four samples (n=4 images) and for the six months’ time point two samples (n=2 images) were quantified. Five regions of interest (ROI) were quantified for each image sample group. This totalizes for the three months’ time point 20 images and for the six months’ time point 10 images”.

4, In figure 4 and figure 6, they y axis value is not readable. The statistical analysis is also not readable. In addition, why there is no quantified results for alpha-SMA?

We thank the reviewer for this comment. We have divided the image panels in two images and increased the graph font to make it readable. We have mentioned at the material and methods section of the manuscript (Pg. 6, lines 276-280) as follow: “Due to regenerated tissue integration with the host over time, the new tissue formed was extremely well integrated making delineation of the scaffold boundaries with native tissue challenge to distinguish, given the absence of fibrous encapsulation. As such to avoid inconsistence while quantifying fibrous tissue encapsulation via α-SMA reactivity, we haven’t quantified αSMA staining, only qualitative analysis was performed”. We have also mentioned in the discussion section (Pg. 16, lines 592-593) of the manuscript that: “in contrast to existing literature [56 - doi:10.1093/rb/rbaa006], our study did not observe fibrotic encapsulation based on the expression of α-SMA”.

Reviewer 3 Report

The current manuscript (bioengineering-2583934) reported their study on the preparation of a medical grade polycaprolactone (mPCL) scaffolds and its potential using for vaginal and abdominal hernia reconstruction with an ovine model. Mechanical properties of the tissue-mesh complexes were showed that, after 6 months of taking surgery, the mPCL mesh was less stiff than the polypropylene (PP) mesh, but stiffer than the native tissue, while showing equitable collagen and vascular ingrowth when compared to PP mesh, which indicated that mPCL is a safe and effective biodegradable scaffold for hernia and vaginal prolapse repair. Overall, this work is interesting and offers important information of tissue reparation. Following are some points need to be cleared before acceptance.

1.     Please explain the reason why using PP as the control group, clearly. As it was already mentioned the disadvantage in introduction, there are many reports of development of the biodegradable materials for tissue reparation, such as cellulose or various gels.

2.     Could you please add the contents about the reason having current design in introduction or discussion section.

Author Response

The authors would like to thank the reviewer for taking the time to review our manuscript. We believe we have addressed all reviewer’s queries and have produced a better version of the manuscript. Please find the detailed responses below.

Reviewer’s queries are in black colour font and authors responses are in red colour font. All changes made within the manuscript have been highlighted in yellow throughout the manuscript text.

Reviewer 3.

The current manuscript (bioengineering-2583934) reported their study on the preparation of a medical grade polycaprolactone (mPCL) scaffolds and its potential using for vaginal and abdominal hernia reconstruction with an ovine model. Mechanical properties of the tissue-mesh complexes were showed that, after 6 months of taking surgery, the mPCL mesh was less stiff than the polypropylene (PP) mesh, but stiffer than the native tissue, while showing equitable collagen and vascular ingrowth when compared to PP mesh, which indicated that mPCL is a safe and effective biodegradable scaffold for hernia and vaginal prolapse repair. Overall, this work is interesting and offers important information of tissue reparation. Following are some points need to be cleared before acceptance.

  1. Please explain the reason why using PP as the control group, clearly. As it was already mentioned the disadvantage in introduction, there are many reports of development of the biodegradable materials for tissue reparation, such as cellulose or various gels.

We thank and appreciate the reviewer's comment. Polypropylene meshes remain one of the most commonly and widely used materials in clinical practices for hernia repairing due to its cost-effectiveness, ease of handling characteristics, and ability to incorporated into the abdominal and vaginal walls. As PP meshes are still commonly employed, we used this proprietary material as a control group to compare with our mPCL mesh scaffold outcomes.

  1. Could you please add the contents about the reason having current design in introduction or discussion section.

We thank the reviewer for this suggestion. As described in our previous response to reviewer 1:

The abdominal and vaginal wall muscles exhibit distinctive non-linear behaviour and fulfill a wide spectrum of mechanical functions due to their anisotropic (different directions) characteristic. These functions encompass the regulation of abdominal and vaginal movement and loading distribution in both horizontal and vertical directions. To more faithfully replicate the native tissue characteristics, particularly these complex biomechanical behaviors, we employed an mPCL mesh scaffold with fibres of both curvy and straight architectures and varied pore sizes (500 µm and 1000 µm) in each direction (horizontal and vertical). The use of curvy fibres provided flexibility to one loading direction, whereas straight fibres provided firmness in the other loading direction, leading to mechanical anisotropy and better tissue compliance. In contrast, polypropylene (PP) meshes, owing to their inherent nonabsorbable and inflexible mechanical properties, primarily aimed to maintain structural integrity and reinforce the weakened tissue site through mechanical tension rather than promoting tissue regeneration.

To enhance clarity, we have included passages from the manuscript where we have elaborated on these points:

Manuscript Results section, Page 6, lines 286-293: “It is known that due to the unique structural organisation of collagens fibres with a wavy pattern, soft tissues are characterized by very low stiffness and high flexibility when tested at low strain levels [44]. However, their stiffness increases significantly with increasing strains as the curvy collagen fibres become taut and they start carrying the applied loads very effectively. Also, soft tissues often exhibit a level of structural and mechanical anisotropy as their collagen fibres are arranged in one prominent direction. Inspired by the collagen architecture of native soft tissues, we designed scaffolds featuring both curvy microfibers and straight fibres”.

Manuscript Results section, Page 7, lines 313-321: “The mechanical testing results (Figure 2) performed on the mesh-tissue complexes at the end of the in vivo study showed that mPCL mesh provides a good reinforcement effect to the implantation area. In contrast to the PP mesh, which is inherently very strong, our initial findings indicated that the new tissue formed within the pores of the mPCL mesh acts as a natural support for the weakened soft tissue. This is expected to reduce the long-term biocompatibility issues associated with conventional permanent meshes. Both PP and mPCL meshes exhibited a non-linear deformation behaviour with a J-shaped force-displacement curve where increasing force values were observed with increasing levels of displacements. This closely resembles the deformation behaviour of native soft tissues”.

Manuscript Discussion section, Page 16, lines 509-515: “By incorporating anisotropic properties into the mesh design, such as variations in fiber orientation or architecture, the mesh can better emulate native tissue characteristics and ultimately enhance the overall success of the implantation process. To achieve anisotropic mechanical properties, the design of mPCL meshes incorporated both curvy and straight fiber architectures. Curvy fibres promote flexibility and straight fibres provide firmness, eventually leading to better mechanical conformity with the underlying native tissue”.

Manuscript Discussion section, Page 16-17, lines 521-527: “The smaller pores and closer distances between mPCL mesh inter-struts, resulting in a higher surface area to volume ratio, facilitated collagen deposition and enhanced early mesh/tissue integration. This, in turn, synergistically provided greater mechanical stability for constructive tissue remodeling at later time points. In contrast, the large PP mesh pores required rapid cell colonization, which, in turn, prompted accelerated collagen synthesis. This increased collagen deposition leads to greater stiffness…”.